# Safety and immunogenicity of heterologous boost immunization with an adenovirus type-5-vectored and protein-subunit-based COVID-19 vaccine (Convidecia/ZF2001): A randomized, observer-blinded, placebo-controlled trial

**Pengfei Jin**[1☯], **Xiling Guo**[1☯], **Wei Chen**[2☯], **Shihua Ma**[3], **Hongxing Pan**[1], **Lianpan Dai**[4], **Pan Du**[5,6], **Lili Wang**[3], **Lairun Jin**[7], **Yin Chen**[1], **Fengjuan Shi**[1], **Jingxian Liu**[1], **Xiaoyu Xu**[5], **Yanan Zhang**[2], **George F. Gao**[4], **Cancan Chen**[2], **Jialu Feng**[8], **Jingxin Li**[1,8,9]*, **Fengcai Zhu**[1,8,9]*

1 NHC Key Laboratory of Enteric Pathogenic Microbiology, Jiangsu Province Center for Disease Control and Prevention, Nanjing, PR China, 2 Anhui Zhifei Longcom Biopharmaceutical, Hefei, PR China, 3 Guanyun County Center for Disease Control and Prevention, Guanyun County, PR China, 4 CAS Key Laboratory of Pathogenic Microbiology and Immunology, Institute of Microbiology, Chinese Academy of Sciences, Beijing, PR China, 5 Vazyme Biotech, Nanjing, PR China, 6 Basic Medical Science School, Zhengzhou University, Zhengzhou, PR China, 7 School of Public Health, Southeast University, Nanjing, PR China, 8 School of Public Health, Nanjing Medical University, Nanjing, PR China, 9 Institute of Global Health and Emergency Pharmacy, China Pharmaceutical University, Nanjing, PR China

☯ These authors contributed equally to this work.
* jingxin42102209@126.com (JL); jszfc@vip.sina.com (FZ)

**Data Availability Statement:** All relevant data are within the manuscript and its Supporting

## Abstract

### Background

Heterologous boost vaccination has been proposed as an option to elicit stronger and broader, or longer-lasting immunity. We assessed the safety and immunogenicity of heterologous immunization with a recombinant adenovirus type-5-vectored Coronavirus Disease 2019 (COVID-19) vaccine (Convidecia, hereafter referred to as CV) and a protein-subunit-based COVID-19 vaccine (ZF2001, hereafter referred to as ZF).

### Methods and findings

We conducted a randomized, observer-blinded, placebo-controlled trial, in which healthy adults aged 18 years or older, who have received 1 dose of Convidecia, with no history of Severe Acute Respiratory Syndrome Coronavirus 2 (SARS-CoV-2) infection, were recruited in Jiangsu, China. Sixty participants were randomly assigned (2:1) to receive either 1 dose of ZF2001 or placebo control (trivalent inactivated influenza vaccine (TIV)) administered at 28 days after priming, and received the third injection with ZF2001 at 5 months, referred to as CV/ZF/ZF (D0-D28-M5) and CV/ZF (D0-M5) regimen, respectively. Sixty participants were randomly assigned (2:1) to receive either 1 dose of ZF2001 or TIV administered at 56

Information files. Individual participant data can be obtained with a request to the Ethics Committee of Jiangsu Provincial Center for Disease Prevention and Control (ec@jscdc.cn).

**Funding:** The work was funded by National Natural Science Foundation of China [grant number 82173584, JL], Jiangsu Provincial Key Research and Development Program [grant number BE2021738, FZ], and Anhui Zhifei Longcom Biopharmaceutical Co., Ltd.(http://www. zhifeishengwu.com/). The funders had no role in study design, data collection and analysis, decision to publish, or preparation of the manuscript.

**Competing interests:** I have read the journal's policy and the authors of this manuscript have the following competing interests: JL reports grants from National Natural Science Foundation of China (grant number 82173584). FZ reports grants from Jiangsu Provincial Key Research and Development Program (grant number BE2021738). LD is an inventor of the patent for the protein subunit vaccine ZF2001. WC, YZ and CC are the employees of Anhui Zhifei Longcom Biopharmaceutical. All other authors declare no competing interest.

**Abbreviations:** CI, confidence interval; COVID-19, Coronavirus Disease 2019; ELISA, enzyme-linked immunosorbent assay; GMFI, geometric mean fold increase; GMT, geometric mean titer; GMR, geometric mean ratio; IgG, immunoglobulin G; IQR, interquartile range; PBMC, peripheral blood mononuclear cell; RBD, receptor-binding domain; SARS-CoV-2, Severe Acute Respiratory Syndrome Coronavirus 2; SD, standard deviation; SFU, spot-forming unit; TIV, trivalent inactivated influenza vaccine; WHO, World Health Organization.

days after priming, and received the third injection with ZF2001 at 6 months, referred to as CV/ZF/ZF (D0-D56-M6) and CV/ZF (D0-M6) regimen, respectively. Participants and investigators were masked to the vaccine received but not to the boosting interval. Primary endpoints were the geometric mean titer (GMT) of neutralizing antibodies against wild-type SARS-CoV-2 and 7-day solicited adverse reactions. The primary analysis was done in the intention-to-treat population. Between April 7, 2021 and May 6, 2021, 120 eligible participants were randomly assigned to receive ZF2001/ZF2001 (n = 40) or TIV/ZF2001 (n = 20) 28 days and 5 months post priming, and receive ZF2001/ZF2001 (n = 40) or TIV/ZF2001 (n = 20) 56 days and 6 months post priming. Of them, 7 participants did not receive the third injection with ZF2001. A total of 26 participants (21.7%) reported solicited adverse reactions within 7 days post boost vaccinations, and all the reported adverse reactions were mild, with 13 (32.5%) in CV/ZF/ZF (D0-D28-M5) regimen, 7 (35.0%) in CV/ZF (D0- M5) regimen, 4 (10.0%) in CV/ZF/ZF (D0-D56-M6) regimen, and 2 (10.0%) in CV/ZF (D0-M6) regimen, respectively. At 14 days post first boost, GMTs of neutralizing antibodies in recipients receiving ZF2001 at 28 days and 56 days post priming were 18.7 (95% CI 13.7 to 25.5) and 25.9 (17.0 to 39.3), respectively, with geometric mean ratios of 2.0 (1.2 to 3.5) and 3.4 (1.8 to 6.4) compared to TIV. GMTs at 14 days after second boost of neutralizing antibodies increased to 107.2 (73.7 to 155.8) in CV/ZF/ZF (D0-D28-M5) regimen and 141.2 (83.4 to 238.8) in CV/ZF/ZF (D0-D56-M6) regimen. Two-dose schedules of CV/ZF (D0-M5) and CV/ZF (D0-M6) induced antibody levels comparable with that elicited by 3-dose schedules, with GMTs of 90.5 (45.6, 179.8) and 94.1 (44.0, 200.9), respectively. Study limitations include the absence of vaccine effectiveness in a real-world setting and current lack of immune persistence data.

## Conclusions

Heterologous boosting with ZF2001 following primary vaccination with Convidecia is more immunogenic than a single dose of Convidecia and is not associated with safety concerns. These results support flexibility in cooperating viral vectored and recombinant protein vaccines.

## Trial registration

Study on Heterologous Prime-boost of Recombinant COVID-19 Vaccine (Ad5 Vector) and RBD-based Protein Subunit Vaccine; ClinicalTrial.gov NCT04833101.

---

## Author summary

### Why was this study done?

- With the waning of antibodies against Severe Acute Respiratory Syndrome Coronavirus 2 (SARS-CoV-2) coinciding with the emergence of the new variants, the effectiveness of Coronavirus Disease 2019 (COVID-19) vaccines has declined over time, necessitating booster vaccinations.

- This study was performed to evaluate whether heterologous immunization with a recombinant adenovirus type-5-vectored COVID-19 vaccine (Convidecia) and a protein-subunit-based COVID-19 vaccine (ZF2001) was safe and immunogenic in healthy adults.

## What did the researchers do and find?

- One hundred twenty participants who have received 1 dose of Convidecia were randomly assigned (2:1) to receive either 1 intramuscular dose of ZF2001 or placebo control (trivalent inactivated influenza vaccine (TIV)), administered at either 28 days or 56 days after priming, and received the third injection with ZF2001 at 4 months after second dose.

- All the reported adverse reactions were mild, and the most common adverse reaction was injection-site pain. Heterologous schedules of ZF2001 following the primary vaccination of Convidecia can induce robust immune responses, particularly with a 5 to 6 months prime-boost interval.

## What do these finding mean?

- These data support the use of heterologous immunization with Convidecia and ZF2001.

## Introduction

Coronavirus Disease 2019 (COVID-19), caused by Severe Acute Respiratory Syndrome Coronavirus 2 (SARS-CoV-2), has severely impacted the world in terms of health, society, and economy [1]. The mass vaccination campaigns are fundamental to reducing the burden of disease and the subsequent economic recovery. As of April 7, 2022, 11.4 billion doses have been administered globally, and 64.7% of the world population has received at least 1 dose of a COVID-19 vaccine, but only 14.8% of people in low-income countries have received at least 1 dose [2]. Currently, national regulatory authorities have granted authorizations for more than 15 COVID-19 vaccines, including 4 adenovirus-based vector vaccines: ChAdOx1 nCoV-19 (AstraZeneca), Ad26.COV 2-S (Janssen), rAd26+rAd5 (Gamaleya), and Ad5 nCoV (CanSino, Convidecia).

Compared with mRNA vaccines and protein-subunit-based vaccines containing novel adjuvants against COVID-19 (e.g., BNT162b2, mRNA-1273, NVXCoV2373), adenovirus-vectored vaccines (e.g., ChAdOx1 nCoV-19, Ad26.COV2-S, and Ad5 nCoV) showed a relatively lower immunogenicity and efficacy against symptomatic disease [3,4]. As the waning of antibodies against SARS-CoV-2 coinciding with the emergence of the new variants, the effectiveness of COVID-19 vaccines has declined over time [4–6], necessitating booster vaccinations. For adenovirus-vectored vaccines, preexisting adenovirus immunity is the biggest obstacle for homologous immunization to overcome. Hence, the Sputnik V vaccine programmer deployed a heterologous prime-boost schedule using Ad26 and Ad5-vectored COVID-19 vaccines, induced robust humoral and cellular responses and showed 91.5% efficacy against COVID-19 [7,8]. Additionally, the occurrence of rare, but severe thrombotic events with thrombocytopenia is another challenge for adenovirus-based vaccines [9–11]. Based on both the concerns of

long-term protective effect and safety, it has been recommended to heterologous immunization with ChAdOx1 nCoV-19 or Ad26.COV 2-S followed by an mRNA vaccine [12,13].

Heterologous regimens have been proposed as an option to elicit stronger and broader, or longer-lasting immunity, which is particularly important for COVID-19 vaccines with moderate vaccine efficacy. The results from clinical trials and real-world studies suggested that heterologous prime-boost vaccination of adenovirus-vectored vaccines (ChAdOx1 nCoV-19 or Ad26.COV 2-S) followed by mRNA vaccines (BNT162b2 or mRNA-1273) induced stronger immune responses, and provided higher effectiveness than homologous ChAdOx1 nCoV-19 vaccination [13,14]. Additionally, the results from Com-COV2 and COV-BOOST trials showed heterologous immunization with ChAdOx1 nCoV-19 and NVXCoV2373 induced both humoral and T-cell immune responses superior to that homologous ChAdOx1 nCoV-19 vaccination [15,16]. Robust data on the safety and immunogenicity of heterologous schedules with different COVID-19 vaccines will help enhance deployment flexibility and improve access to vaccines.

Here, we present the safety and immunogenicity of a heterologous prime-boost vaccination of a recombinant adenovirus type-5-vectored COVID-19 vaccine (Convidecia) followed by a recombinant protein-subunit-based COVID-19 vaccine composed of dimeric receptor-binding domain (RBD) (ZF2001) in healthy adults.

## Methods

### Study design and participants

This study was designed as a randomized, observer-blinded, placebo-controlled trial to access the safety and immunogenicity of a heterologous prime-boost immunization with Convidecia and ZF2001 in Guanyun County, Jiangsu Province, China. The trial protocol was reviewed and approved by the institutional review board of the Jiangsu Provincial Center of Disease Control and Prevention (approval number: JSJK2021-A005-02) and performed in accordance with the Declaration of Helsinki and Good Clinical Practice. This trial was registered with ClinicalTrials.gov NCT04833101. The study protocol, including the CONSORT checklist, can be found in S1 Study Protocol and S1 CONSORT Checklist.

In the original protocol, participants who have received 1 dose of Convidecia were randomly assigned (2:1) to receive either 1 intramuscular dose of ZF2001 or placebo control (trivalent inactivated influenza vaccine (TIV)), administered at either 28 days or 56 days after priming. We made a protocol change to add an additional boost vaccination with ZF2001 at 4 months after first boost dose to further boost the immune responses for all the participants. Four permutations of prime-boost schedule were investigated, including receiving ZF2001/ZF2001 at 28 days and 5 months after priming with Convidecia (referred to as CV/ZF/ZF (D0-D28-M5)), receiving ZF2001 at 5 months after priming (referred to as CV/ZF (D0-M5)), receiving ZF2001/ZF2001 at 56 days and 6 months after priming (referred to as CV/ZF/ZF (D0-D56-M6)), and receiving ZF2001 at 6 months after priming (referred to as CV/ZF (D0-M6)) (Fig 1A).

Participants were healthy adults aged 18 years and above who have received a prime Convidecia vaccination within 28 days before the screening visit. Volunteers with a previous clinical or virologic COVID-19 diagnosis or SARS-CoV-2 infection, diagnosis of an immunocompromising or immunodeficiency disorder, or those who have received immunosuppressive therapy were excluded. Women with positive urine pregnancy test were also excluded from this study. Full details of the inclusion and exclusion criteria could be found in the Supporting information material (S1 Text). All participants provided written informed consent before enrollment.

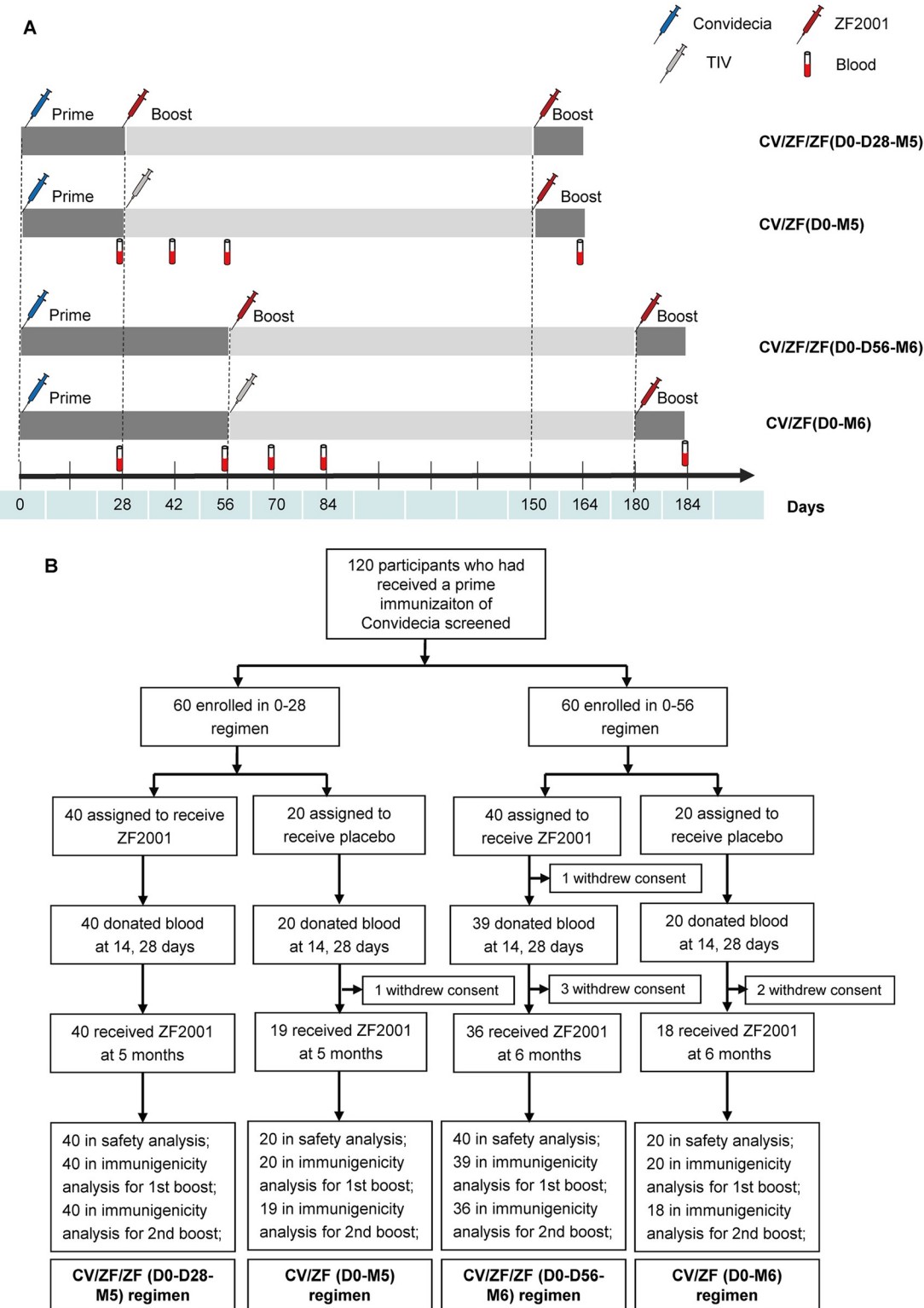

**Fig 1. Study design and trial profile.** (A) Immunization schedule and serum collection in 4 prime-boost regimens. CV/ZF/ZF (D0-D28-M5) refers to receiving Convidecia/ZF2001/ZF2001 at day 0, day 28, and month 5 (day 150); CV/ZF (D0-M5) refers to receiving Convidecia/TIV/ZF2001 at day 0, day 28, and month 5 (day 150); CV/ZF/ZF (D0-D56-M6) refers to receiving Convidecia/ZF2001/ZF2001 at day 0, day 56, and month 6 (day 180); and CV/ZF (D0-M6) refers to receiving Convidecia/TIV/ ZF2001 at day 0, day 56, and month 6 (day 180). For CV/ZF/ZF (D0-D28-M5) and CV/ZF (D0-M5) regimen, blood samples

were collected at day 28 (28 days post priming, before first boost), day 42 (14 days post first boost), day 56 (28 days post first boost), and day 164 (14 days post second boost). For CV/ZF/ZF (D0-D56-M6) and CV/ZF (D0-M6) regimen, blood samples were collected at day 28 (28 days post priming), day 56 (before first boost), day 70 (14 days post first boost), day 84 (28 days post first boost), and day 184 (14 days post second boost). (B) Trial profile. Seven participants discontinued follow up after vaccination. The reasons for dropout are withdrawn consents for participation. TIV, trivalent inactivated influenza vaccine.

## Vaccines

Convidecia and ZF2001 have been authorized for conditional license or emergency use against COVID-19 in China and other countries, and manufactured by CanSino Biologics and Anhui Zhifei Longcom Biopharmaceutical, respectively. ZF2001 encodes the SARS-CoV-2 receptor-binding domain (RBD) antigen (residues 319 to 537, accession number YP_009724390), with 2 copies in tandem repeat dimeric form, and was manufactured in the CHOZN CHO K1 cell line as a liquid formulation containing 25 μg per 0.5 ml in a vial, with aluminium hydroxide as the adjuvant [17]. The control influenza vaccine is produced by Dalian Aleph Biomedical. Administration is via 0.5 ml intramuscular injection into the upper arm for both ZF2001 and TIV.

## Randomization and masking

The randomization lists were generated by an independent statistician using SAS (version 9·4). Participants were stratified by age (18 to 59 years and ≥60 years), and then randomly assigned (2:1) to vaccine group receiving ZF2001 and "placebo" group receiving a TIV either 28 days or 56 days apart. Randomization was done by random block size of 6. Influenza vaccine was used as a placebo comparator rather than a saline placebo to maintain masking of participants and ensure that all participants obtain some benefit.

We masked participants, investigators, laboratory staff, and outcome assessors to the allocation of treatment groups, but not the prime-boost intervals. Personnel who prepared and administered vaccination were aware of group allocation, but they were not otherwise involved in other trial procedures or data collection, and were instructed not to reveal the identity of the study vaccines to the participants and other investigators.

## Procedures

We recruited participants from a population cohort receiving Convidecia, in which participants had no history of SARS-CoV-2 infection before priming with Convidecia, confirmed by SARS-CoV-2 IgM/immunoglobulin G (IgG) antibodies rapid test kit (Vazyme, Nanjing, China). Eligible participants were randomly assigned to receive 1 shot of ZF2001 or TIV in a ratio of 2:1 at 28 days or 56 days after the prime vaccination. Four months after receiving the first boost dose, all the participants were administered with an additional dose of ZF2001. Participants were monitored for 30 minutes post each vaccination for any immediate adverse reactions and instructed to record solicited adverse events up to day 7 and unsolicited adverse events up to day 28 after each dose on paper diary cards. Serious adverse events self-reported by participants were documented throughout the study. Adverse events were graded as mild (grade 1), moderate (grade 2), severe (grade 3), or life-threatening (grade 4) according to the scale issued by the China State Food and Drug Administration (version 2019). Safety data within 28 days after priming with Convidecia were collected in the same manner as the boost vaccination. If a participant developed fever accompanied by respiratory symptoms, he/she was instructed to seek medical attention and notify study staff. For the suspected COVID-19, the participant had a nasal/throat swab taken for PCR test.

Blood samples were taken on day 28 (referred to as baseline), day 42, day 56, day 164, after priming in CV/ZF/ZF (D0-D28-M5) and CV/ ZF (D0-M5) regimens, and on day 28 (referred to as baseline), day 56, day 70, day 84, day 194, after priming in CV/ZF/ZF (D0-D56-M6) and CV/ ZF (D0-M6) regimens (Fig 1A). A microneutralization assay with 50% tissue culture infectious dose of 100 in each well (100 TCID$_{50}$) was used to measure live virus neutralizing antibodies. IgG binding antibody concentrations against RBD and spike protein were detected by an indirect enzyme-linked immunosorbent assay (ELISA) (Vazyme Biotech). The World Health Organization (WHO) international standard for anti-SARS-CoV-2 immunoglobulin (NIBSC code 20/136) was used side by side as reference with the serum samples measured in this study for calibration and harmonization of the serological assays [18]. The WHO reference serum (1,000 IU/ml) is equivalent to neutralizing antibody titer of 1:320 against wild-type SARS-CoV-2. Seroconversion was defined as at least a 4-fold increase in the antibody titers at different time points after priming compared to the baseline level (on day 28 post priming).

Peripheral blood mononuclear cells (PBMCs) were collected at 28 days post prime dose and 14 days post each boost dose, and for the determination of T cells producing interferon (IFN) γ. IFN-γ responses to spike glycoprotein were evaluated using enzyme-linked immunospot (ELISpot) (Mabtech, Stockholm, Sweden) and were expressed as spot-forming units (SFUs) per million PBMCs. An ad hoc analysis was conducted, comparing the wild-type virus neutralizing antibodies to a sample serum panel from 40 participants receiving homologous immunization of Convidecia following a 0 to 56 days regimen and 20 participants receiving homologous prime-boost vaccination of Convidecia with a 0 to 6 months regimen in previous trials [19,20]. Details of assays are included in the Supporting information (S2 Text).

## Outcomes

The primary outcomes were the occurrence of solicited local or systemic adverse reactions within 7 days post vaccination and neutralizing antibody titers against wild-type SARS-CoV-2 at day 14 after each boost vaccination. Safety secondary outcomes include unsolicited adverse events for 28 days after immunization and serious adverse events collected throughout the study. Immunological secondary outcomes include the binding IgG concentration against SARS-CoV-2-specific RBD and spike protein at day 14 after each boost vaccination, and neutralization antibody titers against wild-type SARS-CoV-2 and binding IgG concentration at day 28 post prime dose, at day 28 post first boost dose and at month 6 post second boost dose.

The exploratory outcomes were neutralizing antibodies against the Delta variant B.1.617.2 at 28 days post prime dose and 14 days post second boost dose, and cellular responses measured by IFN γ ELISpot in peripheral blood at 28 days post prime dose and 14 days post each boost dose.

## Statistical analyses

We assumed that the geometric mean titer (GMT) of neutralizing antibodies was about 1:20 at baseline (28 days after 1 dose of prime vaccination with Convidecia). After the second dose, GMT in the vaccine group was expected to reach 1:60 at day 14 post boost vaccination, while that in the control group remained unchanged. Assuming a standard deviation (SD) of 4, 40, and 20 participants receiving vaccine and placebo control in each regimen group, respectively, was estimated to provide 81.6% power for declaring the superiority.

Participants who received at least 1 dose of vaccine were included in the safety analysis. The primary immunogenicity analysis was done in the intention-to-treat population, including all participants who were injected and donated blood samples for antibody tests post boost vaccination. Subgroup analyses were performed according to the age for neutralizing antibodies

against wild-type SARS-CoV-2. We assessed the number and proportion of participants with adverse reactions post vaccination. The antibodies against SARS-CoV-2 were presented as GMTs, geometric mean fold increases (GMFIs), and seroconversion with 95% confidence intervals (CIs), and the cellular responses were shown as the average number of positive cells per PBMCs. We used the $\chi^2$ test or Fisher's exact test to analyze categorical data, and $t$ test to analyze the log-transformed antibody titers. Normal distribution of the data was tested using Kolmogorov–Smirnov test. For nonnormal distributed data, Wilcoxon rank-sum test was used. The correlation between concentrations of log-transformed neutralizing antibodies and binding antibody was analyzed using Spearman's correlation with 95% CIs. Hypothesis testing was 2-sided with an α value of 0.05. Statistical analyses were done by a statistician using SAS (version 9·4) or GraphPad Prism 8.0.1.

## Results

### Study participants

Between April 7, 2021 and May 6, 2021, a total of 120 adults over 18 years of age who had received a primary dose of Convidecia were enrolled, among whom 60 were randomly assigned (2:1) to receive a dose of ZF2001 ($n = 40$) or placebo control (TIV, $n = 20$) at 28 days after priming, and 60 were randomly assigned (2:1) to receive a dose of ZF2001 ($n = 40$) or TIV ($n = 20$) at 56 days after priming. A total of 113 (94.2%) participants received the third injection with ZF2001, with 40 receiving CV/ZF/ZF (D0-D28-M5) regimen, 19 receiving CV/ZF (D0-M5) regimen, 36 receiving CV/ZF/ZF (D0-D56-M5) regimen, and 18 receiving CV/ZF (D0-M6) regimen (Fig 1B). The mean age was 54.0 years (SD 15.0) for the whole study cohort, with 57 (47.5%) female participants. Baseline characteristics of the participants were similar across the 4 regimens (Table 1).

### Safety

A total of 26 participants (21.7%) reported solicited adverse reactions within 7 days post boost vaccination, with 13 (32.5%) in CV/ZF/ZF (D0-D28-M5) regimen, 7 (35.0%) in CV/ZF

Table 1. Baseline characteristics of the participants by vaccination schedules.

| | CV/ZF/ZF (D0-D28-M5) regimen | CV/ZF (D0-M5) regimen | CV/ZF/ZF (D0-D56-M6) regimen | CV/ZF (D0-M6) regimen |
|---|---|---|---|---|
| N | 40 | 20 | 40 | 20 |
| Age, years | 54.6 (15.0) | 51.9 (16.8) | 54.2 (14.5) | 51.6 (15.0) |
| Age group | | | | |
| 18–59 years | 20 (50%) | 10 (50%) | 20 (50%) | 10 (50%) |
| ≥60 years | 20 (50%) | 10 (50%) | 20 (50%) | 10 (50%) |
| Sex | | | | |
| Female | 19 (48%) | 9 (45%) | 20 (50%) | 9 (45%) |
| Male | 21 (53%) | 11 (55%) | 20 (50%) | 11 (55%) |
| Body mass index (kg/m$^2$) | 25.7 (3.3) | 26.0 (2.9) | 24.7 (2.8) | 25.5 (2.3) |
| Underlying diseases | | | | |
| Yes | 7 (18%) | 4 (20%) | 4 (10%) | 2 (10%) |
| No | 33 (82%) | 16 (80%) | 36 (90%) | 18 (90%) |

Data are number of participants (%) or mean (SD). CV/ZF/ZF (D0-D28-M5) = receiving Convidecia/ZF2001/ZF2001 at day 0, day 28, and month 5; CV/ZF (D0-M5) = receiving Convidecia/ZF2001 at day 0 and month 5; CV/ZF/ZF (D0-D56-M6) = receiving Convidecia/ZF2001/ZF2001 at day 0, day 56, and month 6; CV/ZF (D0-M6) = receiving Convidecia/ZF2001 at day 0 and month 6.

**Table 2. Adverse reactions occurred within 7 days and unsolicited adverse events within 28 days post vaccination.**

| | Prime dose (N = 120) | First and second boost dose (N = 120) | | | |
|---|---|---|---|---|---|
| | | CV/ZF/ZF (D0-D28-M5) (N = 40) | CV/ZF (D0-M5) (N = 20) | CV/ZF/ZF (D0-D56-M6) (N = 40) | CV/ZF (D0-M6) (N = 20) |
| **Adverse reactions within 7 days post vaccination** | | | | | |
| Total | 21 (17.5) | 13 (32.5) | 7 (35.0) | 4 (10.0) | 2 (10.0) |
| **Injection-site adverse reactions within 7 days post vaccination** | | | | | |
| Total | 9 (7.5) | 13 (32.5) | 7 (35.0) | 3 (7.5) | 2 (10.0) |
| Pain | 9 (7.5) | 13 (32.5)* | 6 (30.0) | 3 (7.5)* | 2 (10.0) |
| Redness | 0 (0.0) | 0 (0.0) | 1 (5.0) | 0 (0.0) | 0 (0.0) |
| Swelling | 1 (0.8) | 1 (2.5) | 0 (0.0) | 0 (0.0) | 0 (0.0) |
| Induration | 0 (0.0) | 1 (2.5) | 0 (0.0) | 0 (0.0) | 0 (0.0) |
| **Systemic adverse reactions within 7 days post vaccination** | | | | | |
| Total | 15 (12.5) | 1 (2.5) | 0 (0.0) | 1 (2.5) | 0 (0.0) |
| Fever | 10 (8.3) | 0 (0.0) | 0 (0.0) | 1 (2.5) | 0 (0.0) |
| Grade 3 | 2 (1.7) | 0 (0.0) | 0 (0.0) | 0 (0.0) | 0 (0.0) |
| Fatigue | 4 (3.3) | 0 (0.0) | 0 (0.0) | 0 (0.0) | 0 (0.0) |
| Headache | 2 (1.7) | 0 (0.0) | 0 (0.0) | 0 (0.0) | 0 (0.0) |
| Muscle pain | 1 (0.8) | 0 (0.0) | 0 (0.0) | 0 (0.0) | 0 (0.0) |
| Cough | 2 (1.7) | 1 (2.5) | 0 (0.0) | 0 (0.0) | 0 (0.0) |
| **Unsolicited adverse events within 28 days post vaccination** | | | | | |
| Total | 7 (5.8) | 2 (5.0) | 1 (5.0) | 1 (2.5) | 2 (10.0) |

Data are n (%): n = the number of participants, % = proportion of participants; N = the number of participants included in the safety analysis.

*There was significant difference for the incidence of injection-site pain between CV/ZF/ZF (D0-D28-M5) regimen and CV/ZF/ZF (D0-D56-M6) regimen (p = 0.005).

CV/ZF/ZF (D0-D28-M5) = receiving Convidecia/ZF2001/ZF2001 at day 0, day 28, and month 5; CV/ZF (D0-M5) = receiving Convidecia/ZF2001 at of day 0 and month 5; CV/ZF/ZF (D0-D56-M6) = receiving Convidecia/ZF2001/ZF2001 at day 0, day 56, and month 6; CV/ZF (D0-M6) = receiving Convidecia/ZF2001 at day 0 and month 6.

(D0-M5) regimen, 4 (10.0%) in CV/ZF/ZF (D0-D56-M6) regimen, and 2 (10.0%) in CV/ZF (D0-M6) regimen, respectively (Table 2). All the reported adverse reactions post boost vaccination were mild, and the most common adverse reaction was injection-site pain (20.0%, 24/120). The incidence of injection-site pain was higher in CV/ZF/ZF (D0-D28-M5) regimen than that in CV/ZF/ZF (D0-D56-M6) regimen (32.5% versus 7.5%, p = 0.005).

Adverse reactions occurring within 7 days after prime immunization with Convidecia were reported by 17.5% (21/120) of the total participants, with injection-site pain (7.5%), fever (8.3%), and fatigue (3.3%) as the most commonly reported symptoms (Table 2). Two participants had grade 3 fever (axilla temperature ≥38.5°C) after prime vaccination. No serious adverse events and COVID-19 cases were reported throughout the trial.

## Immunogenicity

**Neutralizing antibody responses against wild-type SARS-CoV-2.** The neutralization responses were detectable in 50.8% (61/120) of the participants at 28 days post prime vaccination with Convidecia, with GMT of 8.3 (95% CI 7.0 to 9.7). At 14 days post first boost, GMT of wild-type virus neutralizing antibodies in recipients receiving ZF2001 at 28 days and 56 days post priming was 18.7 (95% CI 13.7 to 25.5) and 25.9 (17.0 to 39.3), respectively, with geometric mean ratios of 2.0 (95% CI 1.2 to 3.5) and 3.4 (1.8 to 6.4) compared to TIV. ZF2001 given as the second boost dose induced comparable neutralizing antibodies to wild-type SARS-CoV-2 in 4 prime-boost regimens, with GMTs at 14 days post boost of 107.2 (73.7 to 155.8) in CV/ZF/ZF (D0-D28-M5) regimen, 90.5 (45.6 to 179.8) in CV/ZF (D0-M5) regimen, 141.2 (83.4 to

238.8) in CV/ZF/ZF (D0-D56-M6) regimen, and 94.1 (44.0 to 200.9) in CV/ZF (D0-M6) regimen, respectively (Fig 2A and 2B and Table C in S1 Data). The GMFIs ranged from 8.9 (95% CI 4.8 to 16.7) in CV/ZF (D0-M6) regimen to 17.3 (11.1 to 26.9) CV/ZF/ZF (D0-D56-M6) regimen when compared to baseline (28 days post prime) (Fig 2C and Table C in S1 Data).

At 14 days post the second boost, the seroconversion of neutralizing antibody titers were observed in 90.0% (36/40) of the participants in CV/ZF/ZF (D0-D28-M5), 89.5% (17/19) in CV/ZF (D0-M5), 91.7% (33/36) in CV/ZF/ZF (D0-D56-M6), and 83.3% (15/18) in CV/ZF (D0-M6) regimen, respectively (Fig 2D and Table C in S1 Data). Homologous immunization with Convidecia at "0 to 56 days" regimen and "0 to 6 months" regimen induced neutralizing antibodies with the GMTs of 32.0 (95% CI 23.7 to 43.0) and 123.6 (82.8, 184.5) at 28 days post boost dose, respectively, which were equivalent to those induced by heterologous boost immunization with ZF2001 (S1 Fig). Neutralizing antibodies 14 days after boost dose were numerically higher in participants aged 18 to 59 years than in those over 60 years across 4 regimens, but no significant difference were found between age subgroups due to small sample size (Fig 3 and Table D in S1 Data).

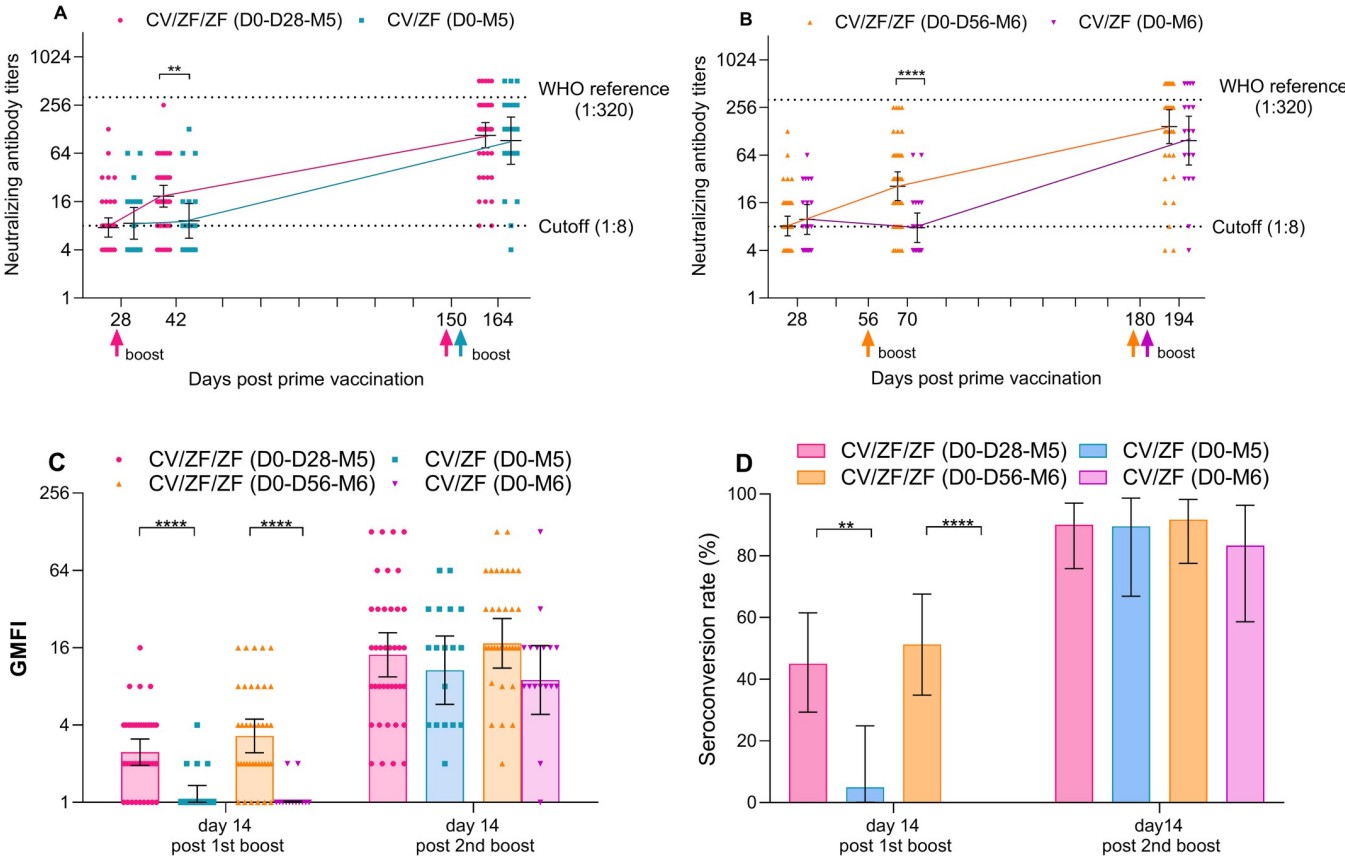

**Fig 2. Neutralizing antibodies against wild-type SARS-CoV-2 after prime and boost dose.** (**A** and **B**) GMTs of neutralizing antibodies to wild-type SARS-CoV-2 28 days after prime dose, 14 days after first and second boost dose (**C**) GMFI of neutralizing antibodies to wild-type SARS-CoV-2 14 days after first and second boost, compared to baseline (28 days post prime). (**D**) Seroconversion rate (%) of neutralizing antibodies to wild-type SARS-CoV-2 14 days after first and second boost. Seroconversion was defined as at least a 4-fold increase in the antibody titers at different time points after boost compared to baseline (28 days post prime). The WHO reference (1,000 IU/ ml) is equivalent to a neutralizing antibody titer of 1:320 against wild-type SARS-CoV-2. Cutoff (1:8) refers to the detection limit. Up arrow represents the boost immunization. Horizontal bars show geometric mean or mean and error bars show 95% CI. **P < 0.05, ****P < 0.001. CV/ZF/ZF (D0-D28-M5) = receiving Convidecia/ZF2001/ZF2001 at day 0, day 28, and month 5; CV/ZF (D0-M5) = receiving Convidecia/ZF2001 at day 0 and month 5; CV/ZF/ZF (D0-D56-M6) = receiving Convidecia/ZF2001/ZF2001 at day 0, day 56, and month 6; CV/ZF (D0-M6) = receiving Convidecia/ZF2001 at day 0 and month 6. CI, confidence interval; GMFI, geometric mean fold increase; GMT, geometric mean titer; SARS-CoV-2, Severe Acute Respiratory Syndrome Coronavirus 2; WHO, World Health Organization.

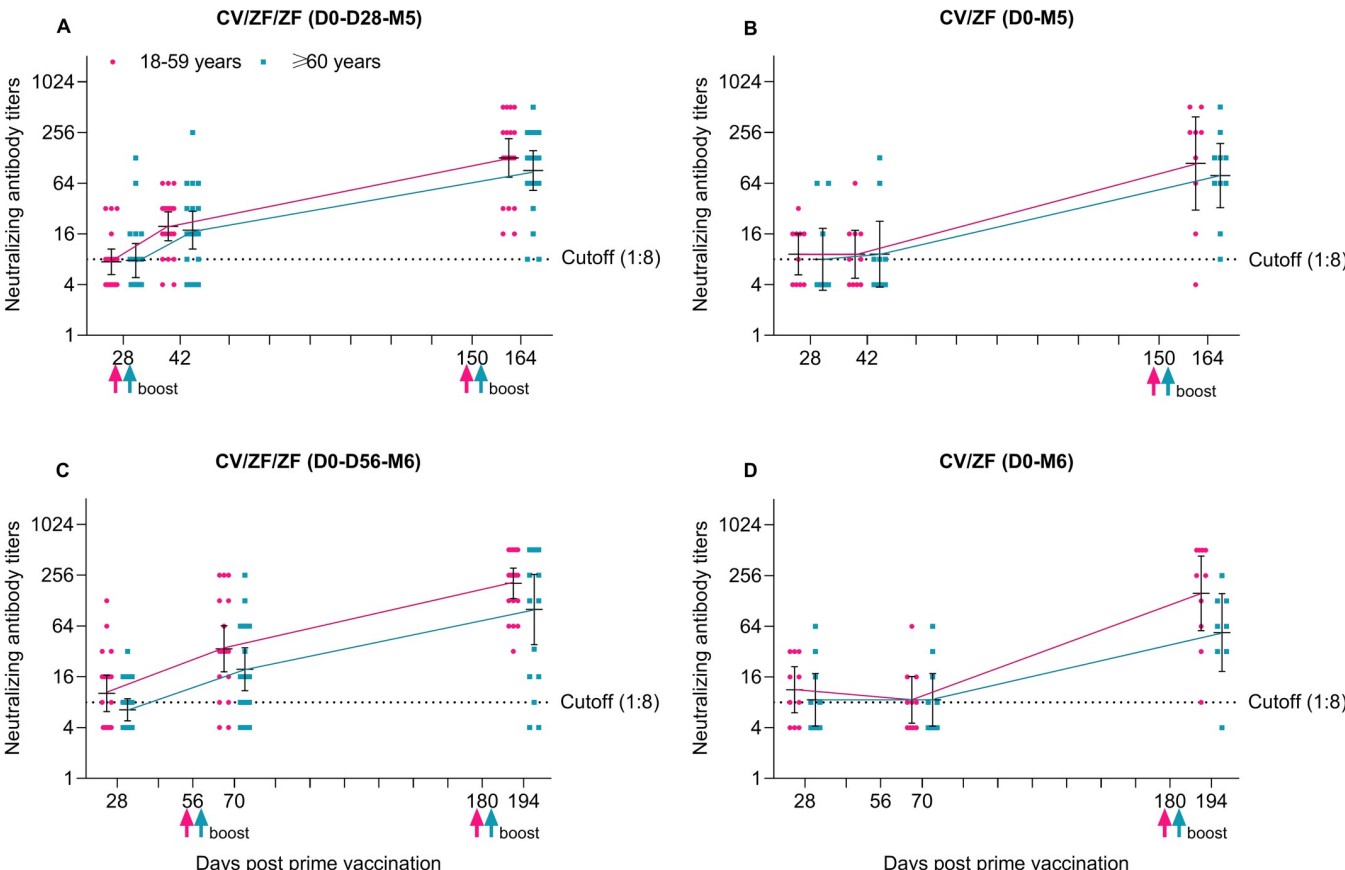

**Fig 3. Neutralizing antibodies to wild-type SARS-CoV-2 after prime and boost dose according to age.** Data presented are neutralizing antibodies to wild-type SARS-CoV-2 28 days after prime dose, 14 days after first and second boost according to age (18–59 years and ≥60 years) in each regimen. Cutoff (1:8) refers to the detection limit. Horizontal bars show geometric mean titer and error bars show 95% CI. Up arrow represents the boost immunization. CV/ZF/ZF (D0-D28-M5) = receiving Convidecia/ZF2001/ZF2001 at day 0, day 28, and month 5; CV/ZF (D0-M5) = receiving Convidecia/ZF2001 at day 0 and month 5; CV/ZF/ZF (D0-D56-M6) = receiving Convidecia/ZF2001/ZF2001 at day 0, day 56, and month 6; CV/ZF (D0-M6) = receiving Convidecia/ZF2001 at day 0 and month 6. CI, confidence interval; SARS-CoV-2, Severe Acute Respiratory Syndrome Coronavirus 2.

## Anti-RBD and anti-spike IgG antibody responses

In line with wild-type virus neutralizing antibodies, ZF2001 induced significantly higher anti-RBD IgG and anti-spike IgG 14 days post first boost, compared with TIV (Fig 4). Anti-RBD IgG Geometric mean ratios (GMRs) between ZF2001 and TIV were 5.2 (2.6, 10.4) in "D0-D28" regimen and 9.3 (4.3, 20.4) in "D0-D56" regimen. GMRs for anti-spike IgG were consistent with those of anti-RBD IgG (Table E in S1 Data). The second boost immunization with ZF2001 increased anti-RBD IgG to comparable level, with GMTs at 14 days post boost of 695.6 IU/ml (95% CI 465.9 to 1,038.5) in CV/ZF/ZF (D0-D28-M5) regimen, 514.7 IU/ml (255.9 to 1,035·2) in CV/ZF (D0-M5) regimen, 951.4 IU/ml (594.0 to 1,523.9) in CV/ZF/ZF (D0-D56-M6) regimen, and 534.5 IU/ml (256.7 to 1,112.9) in CV/ZF (D0-M6) regimen, respectively (Fig 4A and 4B, Table E in S1 Data). Compared with anti-RBD IgG, the GMTs of anti-spike IgG were reduced according to point estimates, with the GMTs of 571.9 IU/ml (95% CI 396.9 to 823.9) in CV/ZF/ZF(D0-D28-M5) regimen, 412.9 IU/ml (202.1 to 843.9) in CF/ZF (D0-M5) regimen, 686.1 IU/ml (435.8 to 1,080.4) in CF/ZF/ZF IU/ml (D0-D56-M6) regimen, and 407.3 IU/ml (211.4 to 784.9) in CF/ZF (D0-M6) regimen, respectively (Fig 4C and 4D and Table E in S1 Data).

Similar patterns of binding antibody responses were observed in age subgroups, and the younger adults had numerically higher humoral responses than did the older adults (S2 and

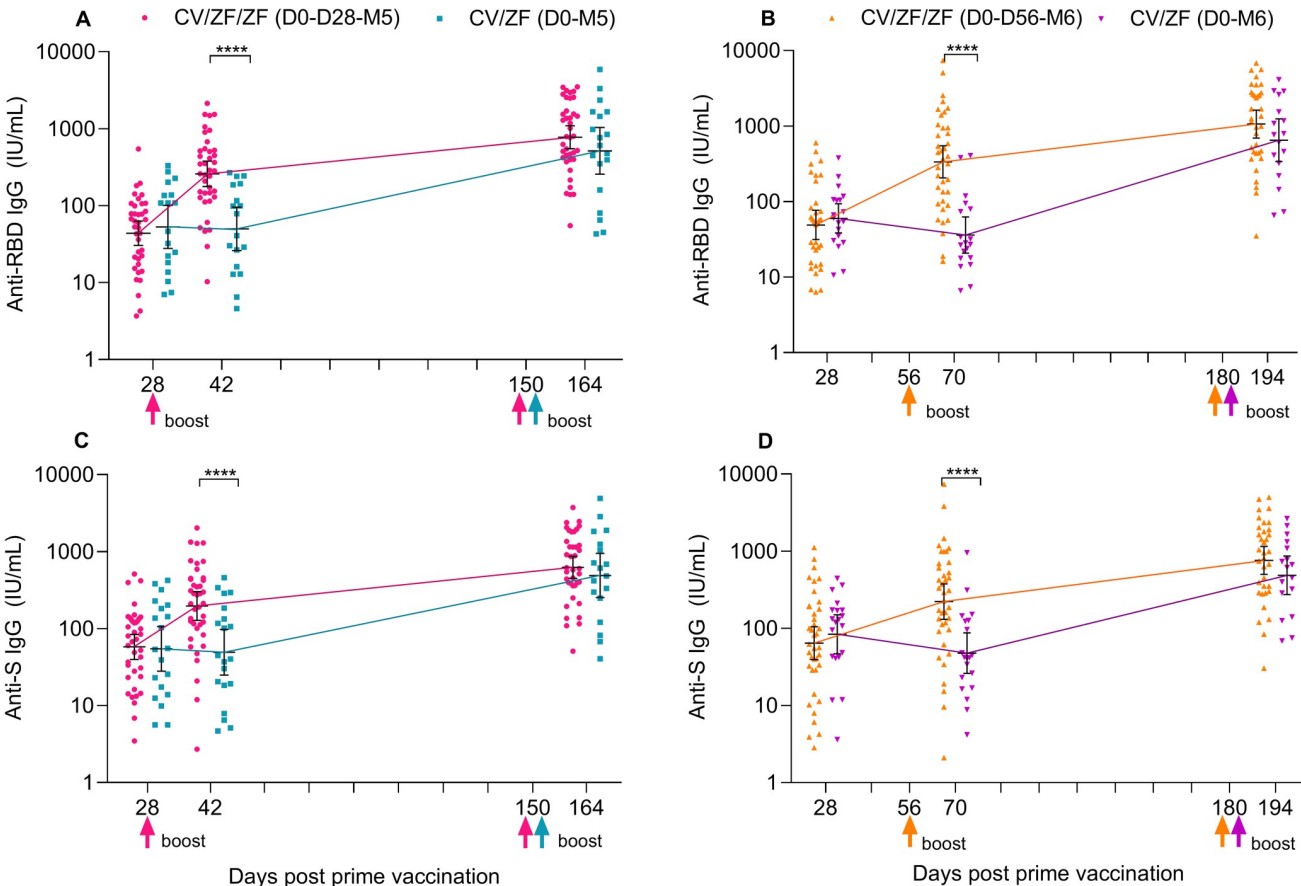

**Fig 4. SARS-CoV-2-specific IgG antibodies after prime and boost dose.** (**A** and **B**) GMTs of anti-RBD IgG (**A** and **B**) and anti-spike IgG (**C** and **D**) 28 days after prime dose, 14 days after first and second boost dose. Horizontal bars show geometric mean concentration and error bars show 95% CI. Up arrow represents the boost immunization. ****$P < 0.001$. RBD = SARS-CoV-2 receptor-binding domain. IU/ml = International units per milliliter. CV/ZF/ZF (D0-D28-M5) = receiving Convidecia/ZF2001/ZF2001 at day 0, day 28, and month 5; CV/ZF (D0-M5) = receiving Convidecia/ZF2001 at day 0 and month 5; CV/ZF/ZF (D0-D56-M6) = receiving Convidecia/ZF2001/ZF2001 at day 0, day 56, and month 6; CV/ZF (D0-M6) = receiving Convidecia/ZF2001 at day 0 and month 6. CI, confidence interval; IgG, immunoglobulin G; RBD, receptor-binding domain; SARS-CoV-2, Severe Acute Respiratory Syndrome Coronavirus 2.

S3 Figs). Strong correlations were found between neutralizing antibodies against wild-type and anti-RBD IgG, and neutralizing antibodies against wild-type and anti-spike IgG at 14 days post second boost dose (Pearson correlation coefficients of 0.87 to 0.92) (S4 Fig).

## Neutralizing antibody responses against SARS-CoV-2 variants

Twenty-eight days after prime immunization with Convidecia, the GMT of neutralizing antibody titers to B.1.617.2 variant was 2.8 (95% CI 2.5 to 3.2). Fourteen days post the second boost vaccination, the GMTs of neutralizing antibody titers to B.1.617.2 variant increased to 38.0 (95% CI 26.7 to 54.2) in CV/ZF/ZF (D0-D28-M5) regimen, 29·7 (15.3 to 57.9) in CV/ZF (D0-M5) regimen, 41.9 (27.0 to 65.0) in CV/ZF/ZF (D0-D56-M6) regimen, and 34.6 (18.1 to 65.9) CV/ZF (D0-M6) regimen, respectively (Fig 5C and Table C in S1 Data). Similar with that after prime immunization with Convidecia, the GMRs of neutralizing antibodies against wild-type relative to Delta variant elicited by boost vaccination ranged 2.9 and 3.4 across 4 heterologous regimens (Fig 5A and 5B). In addition, we tested the serum samples for neutralizing antibodies against the Omicron variant in CV/ZF/ZF (D0-D56-M6) regimen. The results showed

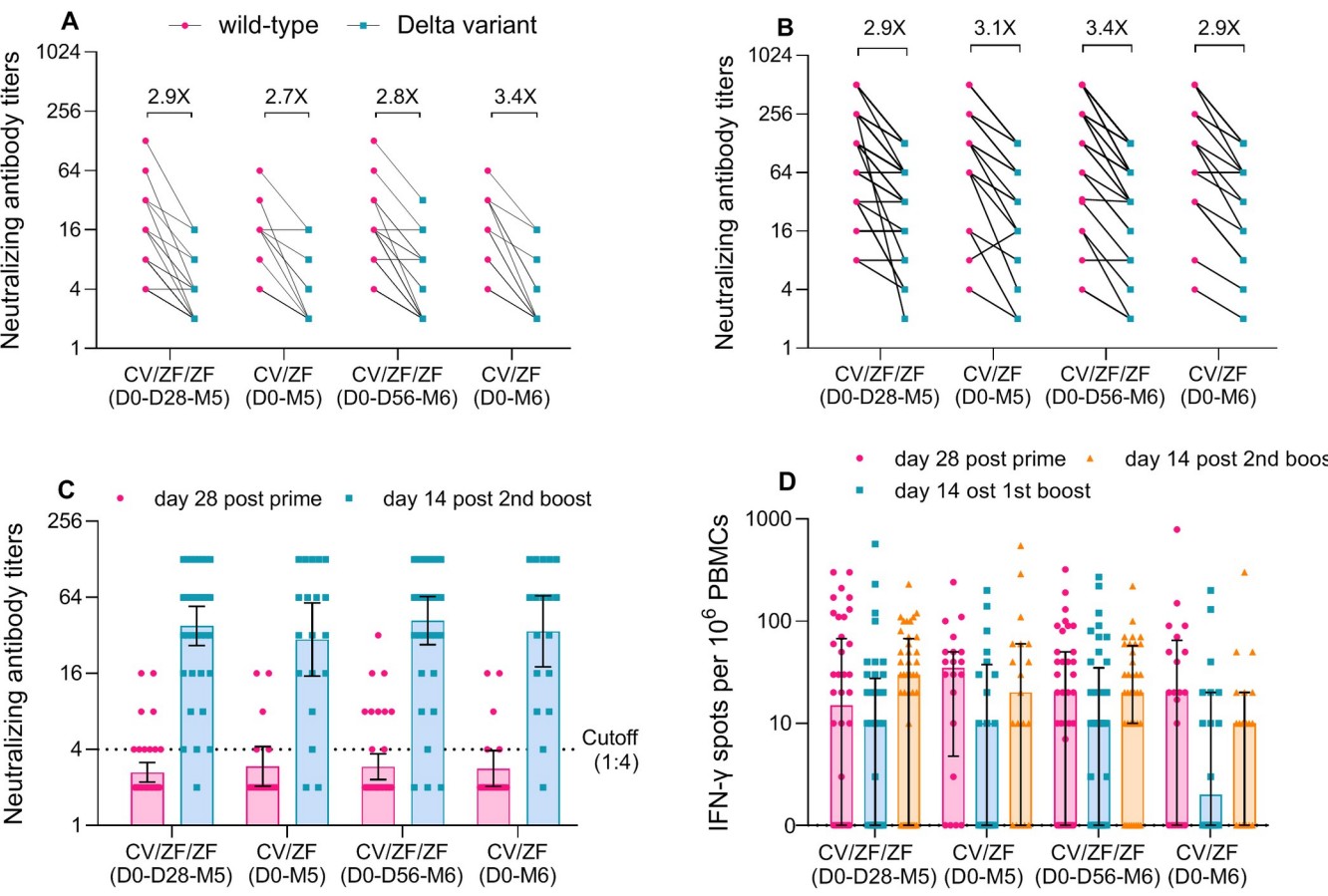

**Fig 5. Neutralizing antibodies against the Delta variant and specific T-cell responses after prime and boost dose. (A** and **B)** Geometric mean ratios of neutralizing antibodies against wild-type relative to Delta 28 days post prime dose (**A**), and 14 days post second boost dose (**B**). (**C**) GMTs of neutralizing antibodies to the Delta variant 28 days post prime dose and 14 days post second boost dose. (**D**) Spot-forming cells with secretion of IFN-γ cytokines per $1 \times 10^6$ PBMCs measured by ELISpot 28 days post prime dose, and 14 days post first and second boost dose. The numbers indicate the geometric mean ratios. Cutoff (1:4) refers to the detection limit. Horizontal bars show geometric mean titer (C) and median (D). Error bars show 95% CI in panel (**C**), and interquartile range in panel (**D**). CV/ZF/ZF (D0-D28-M5) = receiving Convidecia/ZF2001/ZF2001 at day 0, day 28, and month 5; CV/ZF (D0-M5) = receiving Convidecia/ZF2001 at day 0 and month 5; CV/ZF/ZF (D0-D56-M6) = receiving Convidecia/ZF2001/ZF2001 at day 0, day 56, and month 6; CV/ZF (D0-M6) = receiving Convidecia/ZF2001 at day 0 and month 6. CI, confidence interval; ELISpot, enzyme-linked immunospot; GMT, geometric mean titer; IFN, interferon; PBMC, peripheral blood mononuclear cell.

that 8 (27.6%) of 29 serum samples 14 days post first boost and 21 (70.0%) of 30 serum samples 14 days post second boost were positive for neutralizing antibodies against the Omicron variant (BA.1.1). The second boost vaccination induced higher neutralizing antibodies against Omicron than that elicited by first boost, with the GMT of 8.2 (95% CI 5.2 to 12.8) versus 3.6 (2.4 to 5.5) (*p*<0.001). The titer of neutralizing antibodies against the Omicron variant was lower than that against the wild-type SARS-CoV-2 by a factor of 7.8 (95% CI 5.4 to 11.2) 14 days post first boost, and 19.7 (13.6 to 28.6) 14 days post second boost (S5 Fig).

## Vaccine-induced T-cell responses

Ad5-vectored COVID-19 vaccine induced significant specific T-cell responses measured by enzyme-linked immunospot (ELISpot) assay. A median of 20.0 (interquartile range (IQR): 0.0, 57.5) spot-forming cells secreting IFN-γ per $1 \times 10^6$ PBMCs was observed at 28 days after prime vaccination, and no further increase was seen post boost immunization with ZF2001 (Fig 5D). The median of IFN-γ spot counts per $10^6$ PBMCs 14 days after first boost was 10.0

(IQR: 0.0, 27.5) in CV/ZF/ZF (D0-D28-M5) regimen, 10.0 (IQR: 0.0, 37.5) in CV/ZF (D0-M5) regimen, 10.0 (IQR: 0.0, 25.0) in CV/ZF/ZF (D0-D56-M6) regimen, and 1.5 (IQR: 0.0, 20.0) in CV/ZF (D0-M6) regimen, respectively. The median of IFN-γ spot counts per $10^6$ PBMCs 14 days after second boost was 30.0 (IQR: 0.0, 65.0) in CV/ZF/ZF (D0-D28-M5) regimen, 20.0 (IQR: 0.0, 60.0) in CV/ZF (D0-M5) regimen, 20.0 (IQR: 10.0, 55.0) in CV/ZF/ZF (D0-D56-M6) regimen, and 10 (IQR: 0.0, 20.0) in CV/ZF (D0-M6) regimen, respectively.

## Discussion

Our findings show that heterologous immunization of ZF2001 in individuals primed with Convidecia is not associated with safety concerns in this study. The lower frequency of systemic adverse reactions was reported after boost dose with ZF2001, compared with that after prime dose with Convidecia. Four prime-boost regimens induced robust immune responses, with GMTs of wild-type virus neutralizing antibodies ranged 91 to 141, which is equivalent to range 283 to 441 IU/ml using the WHO international standard. This is comparable to 344 IU/ml in participants primed with Janssen Ad26.COV2-S and then given BNT162b2 [21]. Heterologous immunization of ZF2001 with prime-boost intervals of 28 days or 56 days induced 2.5- and 3.3-fold higher neutralizing antibodies against wild-type SARS-CoV-2 than those induced by a single dose of Convidecia, respectively. In addition, we also found that heterologous vaccinations with Convidecia and ZF2001 at an interval of 5 months or 6 months are more efficient in eliciting neutralizing antibodies, with GMFIs of 8.9 and 10.7 compared to a single dose of Convidecia, respectively.

In the present study, we found that the impact of dose interval on the immune responses was greater than the number of doses, which may relate to memory B cell maturation undergoing during 4 to 6 months [22]. Notably, heterologous vaccination with Convidecia and ZF2001 at an interval of 5 months or 6 months induced higher antibody titers than that elicited by immunization at 28 days or 56 days apart. Additionally, 2-dose schedules with D0-M5 and D0-M6 induced antibody level comparable with that elicited by 3 doses of heterologous immunization with D0-D28-M5 and D0-D56-M6 schedules. Zhao and colleagues [23] also showed that prolonged-interval ZF2001 (receiving 3 doses of ZF2001 at interval of month 0, 1, and 4, M0-M1-M4) induced higher binding and neutralizing antibodies than the short-interval ZF2001 (M0-M1-M2), including against SARS-CoV-2 prototype strain and variants of concern such as Delta and Omicron. These findings support the use of a prolonged booster interval to elicit stronger immune responses in persons who had previously received prime immunization of COVID-19 vaccine.

As we know, preexisting anti-adenovirus immunity is the biggest obstacle for the adenovirus-vectored vaccines to overcome, especially for Ad5 eliciting widespread preexisting immunity in the human population. In the previous Phase IIb trial of Convidecia, the boosting effect of homologous prime-boost regime apart 56 days on immune responses was limited due to high anti-Ad5 antibodies [19]. In order to minimize the negative effect of preexisting anti-Ad5 antibody, heterologous prime-boost regimens and a wider prime-boost interval are necessary to provide enhance of immune responses. Our study indicated that homologous Convidecia vaccination apart 6 months induced comparable antibodies with that elicited by heterologous Convidecia-ZF2001 immunization at an interval of 5 months or 6 months. The impact of dose interval was also observed in other vectored COVID-19 vaccines. There is a better immunogenicity when a second dose of Ad26 is given at 6 months after the first dose of Ad26 compared with 2 months [24]. ChAdOx1 nCoV-19 has also shown that a longer prime-boost interval (≥12 weeks) provided higher protective efficacy than a short interval (>6 weeks) [25].

The neutralizing antibodies against the Delta variant elicited by heterologous immunization with Convidecia and ZF2001 decreased about 2.9- to 3.4-fold relative to wild type across the 4

different regimens, and which was similar with that after the prime immunization. Nevertheless, heterologous schedules maintained higher neutralizing antibodies against Delta variant than prime vaccination. In addition, boost vaccination of ZF2001 in individuals primed with Convidecia increased neutralizing antibodies against Omicron and partially restored neutralization of the Omicron variant, but responses were still up to 19-fold decrease compared to wild type. Our results are consistent with those from studies by Servellita and colleagues and GeurtsvanKessel and colleagues and indicate that booster vaccinations are needed to further restore Omicron cross-neutralization by antibodies [26,27]. The use of ZF2001 as a boost does not further increase the cellular immunity responses obtained after the initial dose of Convidecia, which was in line with that reported in a previous trial with ZF2001 booster at interval of 4 to 8 months following 2-dose inactivated vaccines [28]. Compared with protein-subunit-based vaccines containing aluminium adjuvants, those with novel adjuvants could induce stronger immune responses. The results of Com-COV2 study showed that heterologous immunization with ChAdOx1 nCoV-19 vaccine and NVXCoV2373 (a Matrix-M adjuvanted recombinant spike protein vaccine) at an interval of 8 to 12 weeks induced both humoral and T-cell immune responses superior to that elicited by homologous ChAdOx1 nCoV-19 vaccine [15]. The lack of further increase in T-cell responses in our study might be related to the test method, since T cells detected would not continue to increase after reaching a plateau in the ELISpot assay.

Data from the Phase III efficacy trial showed a single dose of Convidecia could provide 57.5% protective efficacy against symptomatic COVID-19 at 28 days or more postvaccination and 91.7% vaccine efficacy against severe disease [29]. The ZF2001 vaccine was shown to be safe and effective against symptomatic and severe-to-critical COVID1-9 for at least 6 months after full vaccination. Three doses (30 days apart) of ZF2001 could provide 75.7% efficacy against symptomatic COVID-19 and 87.6% efficacy against severe-to-critical disease. In addition, vaccine efficacy against Alpha and Delta variants was 88.3% and 76.1%, respectively [30]. Our findings indicate that the heterologous schedule of Convidecia followed by a boost dose of ZF2001 with 5 to 6 months interval increased neutralizing antibodies by 9- to 17-fold, compared with that after an initial dose of Convidecia. Given the established associations between neutralizing antibody titers and vaccine efficacy [3,31], heterologous immunization with Convidecia and ZF2001 5 to 6 months apart are also likely to be highly effective, and could be considered in some circumstances for national vaccine programs.

This study has several limitations. First, the absence of a randomized control group completing the homologous Convidecia scheme. Although we selected 2 extend controls receiving homologous immunization of Convidecia following a 0 to 56 days regimen and 0 to 6 months regimen, which are both comparable with the cohorts receiving heterologous immunization with Convidecia and ZF2001 in baseline characteristics (Table A in S1 Data), there may be some potential bias. As an immunogenicity and reactogenicity study, we do not know whether the immune responses observed in our study will result in better effectiveness, and it is needed to be confirmed in real-world studies. We are unable, at this point, to determine whether higher antibody titers measured at 14 days post boost immunization will result in a more sustained elevation of antibodies, and this will be assessed at 6 months post second boost vaccination. Additionally, we did not collect blood samples before the third dose to assess the true impact of the third dose, and it would explain why there was little difference between 2-dose and 3-dose regimens. Lastly, the sample size of each regimen was relatively small, which limited the statistical power for age subgroup analyses. Findings from this study need to be validated in a large sample size.

In conclusion, our study shows that heterologous schedules of ZF2001 following the primary vaccination of Convidecia are not associated with safety concerns and can induce significant humoral immunity, particularly with a 5 to 6 months prime-boost interval. These results

support flexibility in cooperating viral vectored vaccines and recombinant protein vaccine, subject to supply and logistical considerations.

## Supporting information

**S1 CONSORT checklist. CONSORT 2010 checklist of information to include when reporting a randomised trial.**
(DOCX)

**S1 Study protocol. Study on heterologous prime-boost immunization of recombinant COVID-19 vaccine (Ad5 vector) and RBD-based protein subunit vaccine (CHO).**
(PDF)

**S1 Text. The inclusion and exclusion criteria.**
(DOC)

**S2 Text. Immunogenicity assay method details.**
(DOC)

**S1 Data.** Table A. Baseline characteristics of the participants from external comparators. Table B. Adverse reactions occurred within 7 days and unsolicited adverse events within 28 days post first boost. Table C. Live virus neutralizing antibodies after prime and boost dose. Table D. Wild-type virus neutralizing antibodies after prime and boost dose according to age. Table E. SARS-CoV-2-specific anti-RBD IgG and anti-S IgG antibodies after prime and boost dose.
(DOCX)

**S1 Fig. Neutralizing antibodies against wild-type SARS-CoV-2 day 28 after homologous immunization from external cohorts.** Horizontal bars show geometric mean and error bars show 95% CI. The WHO reference (1,000 IU/ ml) is equivalent to a neutralizing antibody titer of 1:320 against wild-type SARS-CoV-2. Cutoff (1:8) refers to the detection limit. CV/CV (D0-D56) = receiving Convidecia/Convidecia at day 0 and day 56. CV/CV (D0-M6) = receiving Convidecia/Convidecia at day 0 and month 6. CI, confidence interval; SARS-CoV-2, Severe Acute Respiratory Syndrome Coronavirus 2; WHO, World Health Organization.
(TIF)

**S2 Fig. SARS-CoV-2-specific anti-RBD IgG antibodies after prime and boost dose according to age.** Data presented are SARS-CoV-2-specific anti-RBD IgG antibodies 28 days after prime dose, 14 days after first and second boost dose according to age (18–59 years and ≥60 years) in each regimen. Horizontal bars show geometric mean titer and error bars show 95% CI. Up arrow represents the boost immunization. RBD = SARS-CoV-2 receptor-binding domain. IU/ml = International units per milliliter. CV/ZF/ZF (D0-D28-M5) = receiving Convidecia/ZF2001/ZF2001 at day 0, day 28, and month 5; CV/ZF (D0-M5) = receiving Convidecia/ZF2001 at day 0 and month 5; CV/ZF/ZF (D0-D56-M6) = receiving Convidecia/ZF2001/ZF2001 at day 0, day 56, and month 6; CV/ZF (D0-M6) = receiving Convidecia/ZF2001 at day 0 and month 6. CI, confidence interval; IgG, immunoglobulin G; RBD, receptor-binding domain; SARS-CoV-2, Severe Acute Respiratory Syndrome Coronavirus 2.
(TIF)

**S3 Fig. SARS-CoV-2-specific anti-S IgG antibodies after prime and boost dose according to age.** Data presented are SARS-CoV-2-specific anti-spike IgG antibodies 28 days after prime dose, 14 days after first and second boost dose according to age (18–59 years and ≥60 years) in each regimen. Horizontal bars show geometric mean titer and error bars show 95% CI. Up

arrow represents the boost immunization. S = SARS-CoV-2 spike protein. IU/ ml = International units per milliliter. CV/ZF/ZF (D0-D28-M5) = receiving Convidecia/ ZF2001/ZF2001 at day 0, day 28, and month 5; CV/ZF (D0-M5) = receiving Convidecia/ ZF2001 at day 0 and month 5; CV/ZF/ZF (D0-D56-M6) = receiving Convidecia/ZF2001/ ZF2001 at day 0, day 56, and month 6; CV/ZF (D0-M6) = receiving Convidecia/ZF2001 at day 0 and month 6. CI, confidence interval; IgG, immunoglobulin G; SARS-CoV-2, Severe Acute Respiratory Syndrome Coronavirus 2. (TIF)

**S4 Fig. Correlations between immune responses by vaccination schedules.** Correlations at 14 days post second boost were analyzed between neutralizing antibodies to wide-type SARS-- CoV-2 and RBD-specific IgG (A), between neutralizing antibodies to wide-type SARS-CoV-2 and spike-specific IgG (B). Ellipses show the 95% CIs for different vaccine schedules, assuming multivariate normal distributions. Pearson correlation coefficients (95% CIs) are presented for each regimen. CV/ZF/ZF (D0-D28-M5) = receiving Convidecia/ZF2001/ZF2001 at day 0, day 28, and month 5; CV/ZF (D0-M5) = receiving Convidecia/ZF2001 at day 0 and month 5; CV/ ZF/ZF (D0-D56-M6) = receiving Convidecia/ZF2001/ZF2001 at day 0, day 56, and month 6; CV/ZF (D0-M6) = receiving Convidecia/ZF2001 at day 0 and month 6. CI, confidence interval; IgG, immunoglobulin G; RBD, receptor-binding domain; SARS-CoV-2, Severe Acute Respiratory Syndrome Coronavirus 2. (TIF)

**S5 Fig. Neutralizing antibodies against the Omicron variant in CV/ZF/ZF (D0-D56-M6) regimen.** (A) GMTs of neutralizing antibodies against the Omicron variant 14 days post first and second boost. (B) Geometric mean ratios of neutralizing antibodies against wild-type relative to Omicron 14 days post first and second boost. The numbers indicate the geometric mean ratios. Cutoff (1:4) refers to the detection limit. Horizontal bars show geometric mean titer and error bars show 95% CI. CV/ZF/ZF (D0-D56-M6) = receiving Convidecia/ZF2001/ ZF2001 at day 0, day 56, and month 6. ****$P < 0.001$. CI, confidence interval; GMT, geometric mean titer. (TIF)

## Acknowledgments

We thank all study participants enrolled in this trial. We gratefully acknowledge the contributions of other members in our study group: Yongmei Shan, Jiabao Ren, Jing Song, Changyhu Wu, Tongzhou Liu, Shijuan Yan, Jiao Lu, Fanlou Kong, Jing Zheng, Jiahua Jiao, Yongzhe Dai, Yueying Wang, Chunmei Xia, Fuqiang Chen, Qiaoqiao Zheng, and Ping Chen.

## Author Contributions

**Conceptualization:** Pengfei Jin, Wei Chen, Lianpan Dai, George F. Gao, Jingxin Li, Fengcai Zhu.

**Data curation:** Pengfei Jin, Lili Wang.

**Formal analysis:** Lairun Jin, Jialu Feng.

**Funding acquisition:** Wei Chen, Jingxin Li, Fengcai Zhu.

**Investigation:** Pengfei Jin, Xiling Guo, Shihua Ma, Lili Wang, Yin Chen, Fengjuan Shi, Jingxian Liu, Xiaoyu Xu, Jingxin Li.

**Methodology:** Pengfei Jin, Xiling Guo, Pan Du, Jingxin Li.

**Project administration:** Shihua Ma, Hongxing Pan, Jingxin Li.

**Supervision:** Yanan Zhang, Cancan Chen.

**Writing – original draft:** Pengfei Jin, Jingxin Li.

**Writing – review & editing:** Pengfei Jin, Xiling Guo, Wei Chen, Shihua Ma, Hongxing Pan, Lianpan Dai, Pan Du, Lili Wang, Lairun Jin, Yin Chen, Fengjuan Shi, Jingxian Liu, Xiaoyu Xu, Yanan Zhang, George F. Gao, Cancan Chen, Jialu Feng, Jingxin Li, Fengcai Zhu.

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
