## [Editor Report · Decision Letter 0]

23 Feb 2022

Dear Dr Zhu, 

Thank you for submitting your manuscript entitled "Safety and immunogenicity of heterologous boost immunization with an adenovirus type-5-vectored and protein-subunit-based COVID-19 vaccine (Convidecia/ZF2001): a randomized, observer-blinded, placebo-controlled trial" for consideration by PLOS Medicine.

Your manuscript has now been evaluated by the PLOS Medicine editorial staff and I am writing to let you know that we would like to send your submission out for external peer review.

Please re-submit your manuscript within two working days, i.e. by Feb 25 2022 11:59PM.

Kind regards,

Callam Davidson

Associate Editor

PLOS Medicine

---

## [Decision Letter · Decision Letter 1]

25 Mar 2022

Dear Dr. Zhu,

Thank you very much for submitting your manuscript "Safety and immunogenicity of heterologous boost immunization with an adenovirus type-5-vectored and protein-subunit-based COVID-19 vaccine (Convidecia/ZF2001): a randomized, observer-blinded, placebo-controlled trial" (PMEDICINE-D-22-00603R1) for consideration at PLOS Medicine. 

Your paper was evaluated by an associate editor and discussed among all the editors here. It was also discussed with an academic editor with relevant expertise, and sent to independent reviewers, including a statistical reviewer. The reviews are appended at the bottom of this email and any accompanying reviewer attachments can be seen via the link below:

[LINK]

In light of these reviews, I am afraid that we will not be able to accept the manuscript for publication in the journal in its current form, but we would like to consider a revised version that addresses the reviewers' and editors' comments. Obviously we cannot make any decision about publication until we have seen the revised manuscript and your response, and we plan to seek re-review by one or more of the reviewers. 

We hope to receive your revised manuscript by Apr 15 2022 11:59PM. Please email us (plosmedicine@plos.org) if you have any questions or concerns.

We look forward to receiving your revised manuscript. 

Sincerely,

Callam Davidson, 

Associate Editor

PLOS Medicine

plosmedicine.org

Comments from the academic editor:

I think the authors need to include data on omicron responses. Delta is largely irrelevant now, as is the original wuhan variant.

The authors also need to provide a more consistent use of statistics and significance.

Requests from the editor:

The Data Availability Statement (DAS) requires revision: a study author cannot be the contact person for the data. Please direct data requests to a non-author institutional point of contact, such as a data access or ethics committee.

Please report your abstract according to CONSORT for abstracts, following the PLOS Medicine abstract structure (Background, Methods and Findings, Conclusions) http://www.consort-statement.org/extensions?ContentWidgetId=562

Please state that analysis was intention to treat in the Abstract.

At this stage, we ask that you include a short, non-technical Author Summary of your research to make findings accessible to a wide audience that includes both scientists and non-scientists. The Author Summary should immediately follow the Abstract in your revised manuscript. This text is subject to editorial change and should be distinct from the scientific abstract. Please see our author guidelines for more information: https://journals.plos.org/plosmedicine/s/revising-your-manuscript#loc-author-summary.

Please place citations within square brackets throughout the manuscript.

Thanks for providing a CONSORT checklist. When completing the checklist, please use section and paragraph numbers, rather than page numbers.

Please cite your CONSORT checklist in the Methods section (‘This study is reported as per the Consolidated Standards of Reporting Trials (CONSORT) guideline (S1 Checklist)’ or similar).

Similarly, please cite your Protocol/Analysis Plan in the relevant section of the Methods. 

In accordance with ICMJE requirements, PLOS Medicine requires prospective, public registration of a data sharing plan (as part of mandatory clinical trials registration) for all clinical trials that began enrollment on or after January 1, 2019.

The description of the exclusion criteria (Lines 137-141) differ slightly from that presented in Supplementary Table 5 and the clinical trial registry (e.g. previous COVID-19 diagnosis). Please explain this discrepancy.

Please define "lost to follow-up" as used in this study. Other reasons for exclusion should be defined.

Please report the significant difference in injection site pain detailed in the legend of Table 2 in the main text. 

The text in Figures 2-5 is too small to read, please enlarge. 

Consider using a different colour scheme in Figure 2 panels A and B to avoid creating problems for those with red/green colour blindness. 

Throughout, where p values are lower than 0.001, please report as P<0.001. Please also consistently report p values to 3 decimal places. 

Table S3: I could not see the corresponding asterisk for the footnote anywhere in the table.

Throughout the results, when stating a significant difference, please include the quantitative data and relevant p-value (e.g. line 296).

Table S1 is incorrectly labelled Table 1.

Line 373: Please consider tempering the statement ‘is safe’ to ‘was not associated with safety concerns in this study’, or similar. 

Line 437: ‘national vaccine programmes’.

The Author Contributions, Conflict of Interest statement, and Data Availability statements can all be removed from the main text as these are captured as metadata via the Submissions Form. 

Comments from the reviewers:

Reviewer #1: Review of 

"Safety and immunogenicity of heterologous boost immunization with an 2 adenovirus type-5-vectored and protein-subunit-based COVID-19 vaccine 3 (Convidecia/ZF2001): a randomized, observer-blinded, placebo-controlled trial"

In General

This manuscript describes an heterologous boosting study of an adenovirus-based vaccine (Convidecia) and an adjuvanted protein subunit COVID-19 vaccine (ZF2001). Both vaccins are not used in western world countries, but mainly applicated in China, Ecuador, Malaysia, Pakistan, and Uzbekistan. This makes the manuscript less interesting for western countries, however from a scientific viewpoint and from a world perspective, this manuscript has added value.

Introduction

83 This is correct, but AZ is also an adenovirus based platform and its efficacy was also relatively low. The question is whether it is due to only 1 dose or due to the platform? From my point of view, the adeno platforms induce a lower, but longer lasting immune response.

85 With the waning � As the waning….

88 Please let a native english speaker improve the grammar.

94 Based on the both the concerns of � remove the first "the"

Methods

* A figure showing the set up of the protocol would make it easier to read. Or make a new figure (preferred) or refer to figure 1.

123 This is the first time you use the abbreviation TIV.

121 I read this as 28 or 56 days after priming. Is this correct? Please clearify

129 please consider to rephrase these sentences. If you use the words after priming you can state 1 and 5 months after priming.

131 and further.. I would prefer to put the approval in the beginning of the methods section.

138 Sars-cov infection � was this self-reported or also immunological confirmed?

150 The control influenza vaccine � did I miss something? Influenza vaccine? I have not read about this before? Or do you mean placebo?

157 This is the first time I understand that influenza vaccine is used as placebo. Please clarify this earlier. Please adress in the introduction why an influenza vaccine was used and not a regular placebo.

173 grade is missing an e (2x)

176 Please rephrase using less words and making it more simple to read. Please consider to put this is in the figure I was mentioning earlier.

192 Live virus � which variant? I see you mention it in 194. Please try to write more compressed.

201 Unfortunately I am not a virologist. However, I was told (the SARS-CoV-2 virus strain in Vero-E6 cells) that growing SARS COV on vero changes the virus and would not reflect real world data. Please comment on this and consider to put this consideration in the discussion.

198 Please consider to put the Immunogenicity assay-section in the supplemental file.

242 I am not a statistician. This section seems ok for me. But I hope someone else can check as well.

Results

273 Was the external comparator group also mentioned in the protocol submitted? If not, please replace to the discussion section

284 the adverse events after priming were retrospectively determined? Is this stated clearly in the methods section?

288 38.5℃) after prime vaccination. � But I thought that priming was not part of the protocol?

294 Primary endpoint on immunogenicity is 14 days post boost � but you start with reporting 28 days post priming? Which is actually baseline?

298 In the vaccine group � not clear for me which group you mean. Control group?

* Why was no blood drawn on the days of boosting? I cannot see this data in the figures.

* For the whole manuscript I see some sloppy typing errors. E.g in figure 5A Wide type instead of wilde

305-315 this section does not read very easily. Please try to simplify

317 was this comparison with homologous boosting predefined in the protocol?

322 predefined comparison on age?

328 CV/ZF/ZF � CF? see other groups

333 For me it is confusing that you are now looking at 14 days after second boost!!!!

Try to make it easy for the reader to understand. See also 347…. I thought primary endpoint was 14 days post first boost?

345 remove delta � this section is also about wilde type.

Discussion

375 Do you think that 56 days post priming gives less AE than 28 days post priming?

382 "we founded" � we found

416 "not increases the cellular immunity responses " � Tom y opinion this has to do with the lab method (Elispot). In our analysis we also see a plateau in the elispot, but a clear further grow when looked at t-cells with a different method.

@ What happened with participants that became SARS-cov positive during the trial? 

I have not read anything I think

Reviewer #2: Statistical review

This paper reports a randomised trial assessing safety and immunogenicity of heterologous boost vaccination.

Generally the trial is reported well. I have some minor comments.

1. Abstract: Without looking at the main paper, I found it quite hard to follow the different groups in the trial from the abstract. I would recommend emphasising early that there were four groups to help the reader follow the subsequent text.

2. Abstract: as the primary outcome, it might be good to comment on whether there were differences in the adverse reactions between group (or even better, provide the number per group).

3. Abstract: as the trial was powered to detect superiority in GMT of neutralizing antibodies, I would recommend this analysis is summarised in the abstract (preferably with estimated difference, CI and p-value).

4. Randomisation and masking: was stratified block randomisation used?

5. Outcomes: the clinicaltrials.gov page lists some outcomes involving the proportion of patients with at least four-fold increase, which I do not see mentioned in this paper.

6. Results, line 304: it's not clear what these two p-values represent. It would be good to have the estimated difference between arms and CI, either in the text or in a table. Figures 2-4 are good but do not give the differences between arms.

James Wason

Reviewer #3: The manuscript by Jin et al., describes a heterologous boosting strategy for individuals previously vaccinated with one dose of the Adenovirus-vectored vaccine Convidecia, who then received with 1 to 2 doses of the protein subunit vaccine FZ2001. They report the reactogenicity and immunogenicity of the booster doses against ancestral virus and the Delta variant. They show the heterologous primer-boost strategy is safe and well tolerated, and generates a humoral and cellular response.

There are some points that need to be addressed to ensure the manuscript is at the level required for this journal:

Currently there are already a number of commercially available vaccines based on the ancestral Wuhan variant, and numerous reports of heterologous mixing and matching of doses. A current issue in the field is a lack of standardisation across assays to measure neutralising antibody titres. Jin et al., should be commended by addressing this via incorporation of the WHO reference standards, which allow them to report titres in IU/ml. However, despite this there was no real comparison provided to other commercially vaccines which makes it difficult to determine the level of protection this prime-boost strategy would afford, and therefore if it would offer any real benefit compared to other already approved vaccines. In addition, they show that this heterologous prime-boost strategy performs no better (in terms of neutralising antibody titres) than a homologous prime boost with Convidecia at 6 months.

Further analysis to estimate the efficacy of this prime-boost schedule (for example based on the model provided by Khoury et al., Nature Med) would significantly strengthen the manuscript. 

Other studies using mRNA-based vaccines have shown that a 3rd dose at 6-months leads to large increases in neutralising antibody titres compared to pre-boost titres. As pre-3rd dose neutralisation titres were not determined, it is difficult to assess the true impact of the 3rd dose. I.e. Had titres for all individuals had fallen to below detectable levels, and could this explain why there was little difference between the 2-dose versus 3-dose regimen. This information would also provide data on the decay rate of antibodies titres over time.

Inclusion of neutralising antibodies against Omicron would also add to the value of the paper.

Minor points: 

Line 39, should read Convidecia/ZF2001/ZF2001?

Individuals with a previous diagnosis of SARS-CoV-2 were excluded, however were initial serum samples screened for anti- N or M antibodies to confirm none of the individuals had an asymptomatic infection?

In table 2, it is not clear if the adverse reactions are post first or second boost.

Line 298, "in the vaccine group" should be clarified to state which vaccine.

Further information on the ZF2001 would assist the reader, i.e. how is it produced

Reviewer #4: Huge amount of different studies are going on around the world to research COVID-19 vaccines. Today it is obvious that it is necessary to use booster doses of vaccines, especially in the face of the emergence of new variants of the virus.

The authors conducted the study to investigate the safety and immunogenicity of heterologous immunization prime with Ad5-based vector vaccine and boost with RBD-subunit vaccine. The authors provide detailed description of the current state of COVID-19 vaccines development and administration. The provided Protocol of the trial was changed because investigators decided to add additional boost vaccination with protein-subunit vaccine at 4 months after the first boost dose.

As the study was analyzed, a number of questions and comments arose, which are summarized below:

1. The authors investigated several administration regimens of vaccines. One of these schemes included the injection with Convidecia at day 0, the first boost immunization with ZF2001 and the second boost immunization with ZF2001 at 4 months after. This regimen is referred throughout the manuscript as CV/ZF/ZF (D0-D28-M5). However in the abstract it is referred as Convidecia/Convidecia/ZF2001 (Line 39). It should be corrected.

2. Methods and Findings in Abstract - it is not very clear from the description how many groups of vaccinated people there were and in what regimens the vaccines were administered. The authors abbreviate the names of vaccines, while in one place to two letters (lines 40-41), in another to one (lines 55-56). Abbreviations should be brought to uniformity.

3. Lines 113-114 - Authors should add a description of which antigens are included in the subunit vaccine.

4. Line 123 - it is necessary to decipher the abbreviation TIV (decoding is given only on line 157).

5. Line 173 - «grad 1» and «grad 2» - need correction to «grade 1» and «grade 2».

6. Lines 188-193 - The authors describe the immunological studies. The question arises whether the authors analyzed the presence of antibodies specific to the N protein of the SARS-CoV-2 virus in volunteer's sera before the administration of the subunit vaccine and the influenza vaccine. There is no information about this analysis in the protocol. If this analysis was not performed, this should be stated in the study limitations.

7. Line 194 - The authors indicate that the delta variant was used for exploratory outcomes of neutralizing antibodies. Why weren't the results of the analysis of neutralizing antibodies against the wild-type virus (whose glycoprotein homologous to the vaccine) not used for this purpose?

8. Lines 252-256 - The authors give the criteria by which the statistical analysis was carried out. A number of questions arise. What method was used to evaluate the normality of the data distribution in the sample pools? What method was used to evaluate unpaired samples in the case of non-normal distribution? This information should be added to the description of statistical methods.

9. Lines 206-207 and Figure 2 - The authors describe the procedure for the microneutralization reaction, describing that the sera were titrated in two-fold steps from 1/8 to 1/256 (wild-type variant) and from ¼ to 1/128 (Delta variant). The figure also shows the level of neutralizing antibodies in units (not titers), and there is no information about what level of units corresponds to the presence of virus neutralizing antibodies. At the same time, there is no information in the text of the article on how the titer of virus-neutralizing antibodies was converted into units. This information should be added to the immunogenicity analysis section of the methods. There is also confusion when comparing data: the results of the analysis of virus-neutralizing antibodies against the wyld-type variant are given in units, while neutralizing antibodies against the Delta variant are given in titers. In order to limit misunderstanding, it is necessary to bring the results of the analysis to uniformity: either to titers or to units.

10. Figure 5 - it is necessary to make an explanation in the description what the numbers above the brackets mean (0.35, 0.37, etc.)

11. Line 304 - The authors give p-values, but do not explain which data groups they refer to and by what criterion they were analyzed. Authors should add this information.

12. Figure 2 AB, Figure 3, lines 293-323 - in the description of the results it is indicated that not all volunteers had virus-neutralizing antibodies detected. At the same time, when analyzing the figures, a different impression emerges. There is no information in the figures about the level at which units show the presence of virus-neutralizing antibodies, which can confuse the reader.

13. Lines 321-323 - Was the difference in amount of live virus neutralizing antibodies between different age groups statistically significant?

14. Line 357 - Variant B.1.617 indicated. It should be corrected to B.1.617.2.

15. Lines 367-370 - The results of the analysis of the cellular response given in the text do not correspond to the results presented in Figure 5D. The results presented in this place completely copy the results indicated in lines 364-367. Data should be corrected

16. Lines 438-455 - In the limitations of the study, the authors should also indicate the relatively small sample size in some groups (groups of 10 volunteers - Table 1).

General comments:

1. The authors need to check the correctness of the presentation of the results of the study so that the results described in the text correspond to the results presented in the figures.

2. Authors need to bring the results of the study of neutralizing antibodies to uniformity. It is also necessary to indicate the detection limit of neutralizing antibodies in the figures so that the reader does not have a false opinion about the presence of antibodies in the blood serum of all volunteers.

3. In the figures (or as an additional table in the appendix), it would also be useful to indicate the percentage of volunteers with a detectable level of neutralizing antibodies to both variants of the SARS-CoV-2 virus that were used in the study.

In general, the authors present interesting data on the combination of vaccines based on different technological platforms: a vector vaccine and a subunit vaccine. The article can be accepted for publication after making corrections and eliminating inconsistencies.

[LINK]

---

## [Decision Letter · Decision Letter 2]

9 May 2022

Dear Dr. Zhu,

Thank you very much for re-submitting your manuscript "Safety and immunogenicity of heterologous boost immunization with an adenovirus type-5-vectored and protein-subunit-based COVID-19 vaccine (Convidecia/ZF2001): a randomized, observer-blinded, placebo-controlled trial" (PMEDICINE-D-22-00603R2) for review by PLOS Medicine.

I have discussed the paper with my colleagues and the academic editor and it was also seen again by three reviewers. I am pleased to say that provided the remaining editorial and production issues are dealt with we are planning to accept the paper for publication in the journal.

[LINK]

We look forward to receiving the revised manuscript by May 16 2022 11:59PM.   

Sincerely,

Callam Davidson, 

Associate Editor 

PLOS Medicine

plosmedicine.org

Requests from Editors:

Your Data Availability Statement notes that individual participant data will be available for request one month after study completion – please can you provide an update on this and update the DAS if necessary?

Line 35: ‘We conducted…’ 

Lines 37 and 41: ‘Sixty subjects were…’

Line 58: ‘…were 18.7’

Line 63: ‘…induced antibody levels comparable with…’

Line 66: ‘…in a real-world setting…’

Line 68-70, I would propose rephrasing as follows: ‘Heterologous boosting with ZF001 following primary vaccination with Convidecia is more immunogenic than a single dose of Convidecia and is not associated with safety concerns. These results support flexibility in cooperating viral vectored and recombinant protein vaccines.’

Line 111: ‘…coinciding…’

Line 112: ‘…the effectiveness of COVID-19 vaccines has declined over time, necessitating booster vaccinations.’

Line 131: Please either remove ‘in low- and middle-income countries’ or provide further justification for this specific part of the sentence (income levels are not mentioned until this point).

Lines 285-288: Please ensure methods are reported in the past tense (e.g., ‘developed’, ‘was’, ‘had’).

Line 498: ‘…bars…’

Line 712: The original phrasing here is better in my opinion (‘First, the absence of a randomized control group…’)

Line 713: ‘Although we selected two extended controls…’

Line 727: ‘Findings from this study need to be validated…’

Line 736: Please replaces safe with ‘not associated with safety concerns’

Line 740: As with the Author Summary, I feel this line requires further justification at an earlier point in the manuscript (or removal), as low and middle-income countries are not mentioned in any depth until the concluding paragraph. 

To help us extend the reach of your research, please provide any Twitter handle(s) that would be appropriate to tag, including your own, your coauthors’, your institution, funder, or lab. Please respond to this email with any handles you wish to be included when we tweet this paper.

Comments from Reviewers:

Reviewer #2: Thank you to the authors for addressing my previous comments well. I have no further issues to raise.

Reviewer #3: The authors have addresses my previous concerns, however, the manuscript still needs editing to improve the grammar/correct typos.

For example: 

Line 121, thrombocytopenia is other challenge, should read "thrombocytopenia is another challenge"

Line 208 Personals; should be personnel 

The tense needs to be changed in lines 228-231 i.e "if a subject develops" should be if a subject developed etc.

When referring to when the blood samples were taken (lines 232 onwards) it would be helpful to refer back to figure 1 in the text.

Line 265 should say wild-type SARS-CoV-2

Line 506 "As we known" should read as we know

Line 531 ZF2001 as a boost dose not; should read does not

In supplementary text 2 accession numbers/details for the omicron variant are missing.

Reviewer #4: The authors made adjustments according to the comments received. When reviewing the article, one comment remained:

In the background, it is necessary to make a transcript in lines 32 and 33 - Convidencia (CV) and ZF2001 (ZF).

---

## [Editor Report · Decision Letter 3]

12 May 2022

Dear Dr Zhu, 

On behalf of my colleagues and the Academic Editor, Dr James Beeson, I am pleased to inform you that we have agreed to publish your manuscript "Safety and immunogenicity of heterologous boost immunization with an adenovirus type-5-vectored and protein-subunit-based COVID-19 vaccine (Convidecia/ZF2001): a randomized, observer-blinded, placebo-controlled trial" (PMEDICINE-D-22-00603R3) in PLOS Medicine.

When making the formatting changes, please also update your Data Availability Statement to include the full name of the Ethics Committee. 

PRESS

Sincerely, 

Callam Davidson 

Associate Editor 

PLOS Medicine